

# Changes in metabolic profiling of sugarcane leaves induced by endophytic diazotrophic bacteria and humic acids

Natalia O. Aguiar[1], Fabio L. Olivares[1], Etelvino H. Novotny[2] and Luciano P. Canellas[1]

[1] Núcleo de Desenvolvimento de Insumos Biológicos para a Agricultura (NUDIBA), Universidade Estadual do Norte Fluminense, Campos dos Goytacaes, Rio de Janeiro, Brazil

[2] Embrapa Solos, Rio de Janeiro, Rio de Janeiro, Brazil

Corresponding author
Luciano P. Canellas, canellas@uenf.br

## ABSTRACT

Plant growth-promoting bacteria (PGPB) and humic acids (HA) have been used as biostimulants in field conditions. The complete genomic and proteomic transcription of *Herbaspirillum seropedicae* and *Gluconacetobacter diazotrophicus* is available but interpreting and utilizing this information in the field to increase crop performance is challenging. The identification and characterization of metabolites that are induced by genomic changes may be used to improve plant responses to inoculation. The objective of this study was to describe changes in sugarcane metabolic profile that occur when HA and PGPB are used as biostimulants. Inoculum was applied to soil containing 45-day old sugarcane stalks. One week after inoculation, the methanolic extracts from leaves were obtained and analyzed by gas chromatography coupled to time-of-flight mass spectrometry; a total of 1,880 compounds were observed and 280 were identified in all samples. The application of HA significantly decreased the concentration of 15 metabolites, which generally included amino acids. HA increased the levels of 40 compounds, and these included metabolites linked to the stress response (shikimic, caffeic, hydroxycinnamic acids, putrescine, behenic acid, quinoline xylulose, galactose, lactose proline, oxyproline and valeric acid) and cellular growth (adenine and adenosine derivatives, ribose, ribonic acid and citric acid). Similarly, PGPB enhanced the level of metabolites identified in HA-treated soils; e.g., 48 metabolites were elevated and included amino acids, nucleic acids, organic acids, and lipids. Co-inoculation (HA+PGPB) boosted the level of 110 metabolites with respect to non-inoculated controls; these included amino acids, lipids and nitrogenous compounds. Changes in the metabolic profile induced by HA+PGPB influenced both glucose and pentose pathways and resulted in the accumulation of heptuloses and riboses, which are substrates in the nucleoside biosynthesis and shikimic acid pathways. The mevalonate pathway was also activated, thus increasing phytosterol synthesis. The improvement in cellular metabolism observed with PGPB+HA was compatible with high levels of vitamins. Glucuronate and amino sugars were stimulated in addition to the products and intermediary compounds of tricarboxylic acid metabolism. Lipids and amino acids were the main compounds induced by co-inoculation in addition to antioxidants, stress-related metabolites, and compounds involved in cellular redox. The primary compounds observed in each treatment were identified, and the effect of co-inoculation (HA+PGPB) on metabolite levels was discussed.

## INTRODUCTION

Plant inoculation technology is a secure and environmentally-friendly technology that saves approximately 3 billion dollars per year in Brazil by decreasing the application of nitrogen fertilizers in soybean and others leguminous plants (*Alves, Boddey & Urquiaga, 2003*). This technology has considerable potential for non-leguminous plants (*Reis et al., 2008*). Current scientific efforts are focused on the isolation and selection of plant-beneficial microorganisms and their application in soil-plant systems in controlled conditions (*Vassilev et al., 2015*).

Despite the genomic sequencing efforts of *Herbaspirillum seropedicae* and *Gluconacetobacter diazotrophicus* (*Bertalan et al., 2009*; *Pedrosa et al., 2011*), the field responses of sugarcane inoculation with plant growth-promoting bacteria (PGPB) are far from those results obtained in experimental conditions (*Oliveira et al., 2009*; *Carvalho et al., 2014*; *Schultz et al., 2014*). *Da Silva, Olivares & Canellas (2017)* applied a suspension containing humic acids (HA) and PGPB on sugarcane foliage and verified a 37% increase in productivity (26 tons ha$^{-1}$) during the first year as compared to the control. The positive growth promotion in sugarcane co-inoculated with HA and *H. seropedicae* persisted for three years (sugarcane plant and two ratoons). Furthermore, an experiment utilizing large pots containing inoculated sugarcane (emulating commercial plantations) showed larger yields than non-inoculated plants for three consecutive years, further proving the benefits of co-inoculating with HA and PGPB.

Field efficiency is a key aspect for bioinoculant acceptance and wider agricultural use (*Bhattacharyya & Jha, 2012*; *Owen et al., 2015*). *Olivares et al. (2017)* reviewed the use of HA and PGPB on non-leguminous plants and showed different morphological adaptations and physiological changes induced by co-inoculation. The study of plant metabolite levels has contributed significantly to our understanding of plant physiology, particularly from the viewpoint of small chemical molecules that reflect the end point of biological pathways (*Hong et al., 2016*). Changes in the metabolic profile of plants treated with inoculants may further decrease the gap between field and laboratory results. Furthermore, the recognition of cell metabolites altered by inoculation is fundamental for the identification of specific molecular targets. Therefore, the aim of this work was to characterize the changes in the metabolic profile of sugarcane co-inoculated with HA and *H. seropedicae* and *G. diazotrophicus*.

## MATERIALS AND METHODS

### Humic acids (HA) and plant growth-promoting bacteria (PGPB)

Humic substances were extracted and purified as reported elsewhere (*Aguiar et al., 2013*). Briefly, 10 volumes of 0.5 mol L$^{-1}$ NaOH were mixed with one volume of earthworm compost, under a N$_2$ atmosphere and shaken overnight. After 12 h, the suspension was

centrifuged at 5,000 g and acidified to a pH of 1.5 using 6 mol $L^{-1}$ HCl. The HAs were solubilized in 0.5 mol $L^{-1}$ NaOH and precipitated three times. The sample was repeatedly washed with water until a negative test against $AgNO_3$. The HAs were titrated to a pH of 7.0 with 0.1 mol $L^{-1}$ KOH then dialyzed against deionized water using a 1 kD cutoff membrane (Thomas Scientific, Swedesboro, NJ, USA) before they were lyophilized.

Mixed inoculum containing *Gluconacetobacter diazotrophicus* strain PAL5 and *Herbaspirillum seropedicae* strain HRC54 was used in this experiment, both originally isolated from sugarcane in Brazil and characterized as endophytic diazotrophs (*Cavalcante & Dobereiner, 1988*; *Olivares et al., 1996*), and being part of the five-species inoculant recommended for sugarcane in Brazil (*Schultz et al., 2016*). For pre-inoculum preparation, both bacteria species were grown in 5 mL liquid DIGYS medium at 30 °C for 36 h at 140 rpm in rotatory shaker (*Baldani et al., 2014*). Later, 50 µL of each bacterial species was inoculated in separated in 500 mL erlenmeyer flasks containing freshly liquid DIGYS at the same growing condition for 48 h. The bacteria biomass produced was centrifuged at 5.000 g for 10 min and resuspended in sterilized water, being adjusted for a cell density of $10^8$ cells $mL^{-1}$ that correspond to O.D. of 1.0 at 440 ŋ m. The inoculant was prepared by diluting 100 mL of each bacterial suspension in 800 mL of distilled water (bacteria treatment) or in 800 mL humic acid solution at pH 6.5 (bacteria + humic acid treatment), as well 200 mL of distilled water plus 800 mL of humic acid solution (humic acid treatment) and 1,000 ml of distilled water (control treatment).

## Plant assay

Before the planting, we perform the heat treatment and cane stalks were immersed water (50 °C) by 2 h. The sugarcane cultivar RB 96 7515 was used for pot experiments. We use the superficial layer (0–20 cm) of a typical soil used to sugarcane plantation in Rio de Janeiro state and classified as Inceptisol. The soil chemical properties were analyzed according to Embrapa methods described in *Claessen et al. (1997)*: pH ($H_2O$) = 5.5; Al = 3 $mmol_c$ $dm^{-3}$ (titration against NaOH); Ca = 7 $mmol_c dm^{-3}$ and Mg = 4 $mmol_c$ $dm^{-3}$ (titration against EDTA); P = 4.0 mg $dm^{-3}$ (Mehlich 1 extraction and colorimetric determination); K = 33 mg $dm^{-3}$ (Mehlich 1 extraction and flame photometric determination). The randomized complete block design (RCBD) was used as experimental design with four treatments and three replicates. The treatments consisted of a final bacteria suspension at $4 \times 10^8$ cell $mL^{-1}$, humic substances at 50 mg C $L^{-1}$ or its combination which were applied at the same time directly in the soil at 45 days after germination leaf bud at rate of 400 mL per pot. Control treatment was performed applying water at the same volume. The plants were irrigated to keep soil moisture at field capacity. The leaves were collected after 7 days of treatments application to metabolomic profile analyze.

## Sugarcane sample processing and metabolite extraction

Sugarcane leaves were collected under controlled conditions, ground in the $N_2$ homogenizer and stored at −80 °C. Metabolites were extracted using 20 mg of fresh weight sample adding 1 mL of pre-chilled extraction solution (80:20 v/v solvent mixture of methanol/water). The samples were vortex for 10s and shake for 6 min followed by centrifugation for 10 min

at 10,000 g. We remove the whole supernatant in two 500 µL portions, saving one as a backup. One portion was dried and submitted to derivatization.

## GC–TOF mass spectrometry

Derivatization was performed as described previously (*Fiehn et al., 2008*). In summary, 2 µL of a C8-C30 FAME mixture was used to convert retention times to retention index (RI). Carbonyl groups were protected by 10 µL of a solution of 20 mg mL$^{-1}$ methoxyamine in pyridine at 30 °C for 90 min. Ninety microliters of MSTFA and 1% TMCS was added for trimethylsilylation of acidic protons at 37 °C for 30 min. After derivatization 0.5 µL samples were injected in randomized sequence into a Gerstel cold injection system (Gerstel, Muehlheim, Germany) and Agilent 7890A gas chromatograph (Santa Clara, CA, USA) in split less mode. The system was controlled by the Leco ChromaTOF software versus 2.32 (St. Joseph, MI, USA). A 30 m long, 0.25 mm i.d. RTX 5Sil-MS column with 0.25 lm 5% diphenyl/ 95% dimethyl polysiloxane film and additional 10 m integrated guard column was used (Restek, Bellefonte, PA, USA). Injection temperature was 230 °C, the interface was set to 280 °C. Helium flow was 1 mL min$^{-1}$. After a 5 min solvent delay time at 50 °C, oven temperature was increased at 20 °C min$^{-1}$ to 330 °C, 5 min isocratic, cool down to 50 °C and additional 5 min delay, afterwards. Liners were exchanged automatically every 10 samples. A Leco Pegasus IV time-of-flight mass spectrometer was operated at a transfer line temperature of 280 °C, ion source was adjusted at 250 °C and −70 V electron impact ionization. Mass spectra were acquired at mass resolving power $R = 600$ from m/z 85 to 500 at 17 spectra s$^{-1}$. The results files were exported to a data server with absolute spectra intensities and further processed by a filtering algorithm implemented in the metabolomics Bin-Base database (*Fiehn, Wohlgemuth & Scholz, 2005*).

Quantification was reported as peak height using the unique ion as default, unless a different quantification ion was manually set in the BinBase administration software Bellerophon. Metabolites were unambiguously assigned by the Bin-Base identifier numbers, using retention index and mass spectrum as the two most important identification criteria. All database entries in BinBase were matched against the Fiehn mass spectral library of 1,200 authentic metabolite spectra and the NIST05 commercial library (*Kind et al., 2009*). A quantification report table was produced for all database entries that were positively detected in more than 50% of the samples of a study design class as defined in the SetupX database (*Scholz & Fiehn, 2007*).

## Multivariate analysis

The metabolite data were then normalized based on the cell dry weight in each sample. The resulting data sets, which comprised 12 samples and 285 variables, were imported into The Unscramble software (version 10.1; Camo Co., Oslo, Norway) for multivariate statistical and analysis (*Fiehn et al., 2008*; *Lee & Fiehn, 2008*) and univariate analysis (Anova and Dunnet mean test against the control).

## RESULTS

We found 1,885 different compounds in sugarcane leaves extracts and 280 were found in all samples and with accurate identification according to the similarity to chemical standard

of the Fiehn library used in this work (*Lee & Fiehn, 2008*). The identified compounds were mainly of carbohydrates (30%), organic acids (16%), lipids (17%), amino acids (9%), aromatics (11%), others nitrogenous compounds besides amino acids (12%) and other compounds (2%). Identified compounds were sorted according its chemical classification. The values represent percentage difference of the concentration in relation to the control according to the significance analysis of the variance (Tables 1–6). In general, humic acid (HA treatment) induced a decreased amount of larger number of compounds in respect to the control, except for aromatics compounds. For bacteria inoculation (PGPB treatment), remarkable increases of amino acid metabolism and nitrogenous biosynthesis related to control plants were observed. The combined application (HA+PGPB treatment) displayed the highest number of compounds related to the control (Fig. 1). Hereafter, we expose the main results according the biochemical category of leaves metabolites.

## Carbohydrates (Table 1)

All treatments induce the concentration of these sugars: glucose, glucose-1-P, galactose, levanbiose, lactobionic acid, mannose, hexopyranose, deoxyerythriol and sialicin (amino sugar). In the HA treatments we found nine carbohydrates that have its concentration reduced in respect to control. With the inoculation this number was reduced to six and PGPB+HA to zero. The main compounds decreased by HA were: 1,5 anhydrous gluciol (−80%), leucrose (−66%), two generic disaccharides described by numbers 200,391 and 553,367 in the Fiehn library (−36 and −52%, respectively), melibiose (−61%) and fucose (−60%). sucrose-6-phosphate (−42%). The sugar decreased by inoculation were 2-ketogulonic acid, the carbohydrate 208,845 and the raffinose (−50%). The carbohydrates founded in larger concentration with HA treatment were raffinose and ribose (548%), lactose (+510%), xylulose (+458%), salicin (+211%), among others with less increase. The sugars enhanced by bacteria inoculation were erythrol lactone, gluco heptulose, melibiose, 1-deoxy pentiol and two disaccharides (267,647 and 285,065). The PGPB+HA enhance the concentration of ribito (+825%), isomaltose (+586%), melezitose (+533%), glucose-6-phosphate (+413%), sophorose (+297%), glycerol-gluco heptose (+258%), enolpyruvate (+246%), tagalose (+220%), hexose amino-2-deoxy (192%), erythrol (+169%), trehalose (+133%), galactose-6-P (+107%) and fucose (+44%). It was notable the increase of melibiose with inoculation enhancing from 943% to 4,143% when the inoculation was done in the presence of HA.

## Amino acids (Table 2)

Twenty-six amino acids were identified, and the HA reduce the concentration of two including asparagine (−85%) and glutamine (−72%) and enhance significantly lysine (+306%), 5-aminovaleric acid (+211%), isoleucine (+45%), proline (+278%) and oxoproline (+26%). The PGPB and PGPB+HA enhanced the amino acid metabolism increasing mostly including aspartic acid (from 122 to 629%), asparagine (177 to 937%), glutamic acid (254%), glutamine (1,038% to 595%), homoserine (216 to 640%), isoleucine (88 to 135%), N-methyl alanine (100%), ornithine (247% to 425%), hydroxyproline (94 to 369%), proline (1,031 to 1,155%), oxoproline (192 to 320%), citrulline (480 and 446%), lysine (429% to 677%) and methionine (208 to 264%).
**Table 1 Carbohydrates identified on sugarcane methanolic leaves extracts by GC-MS TOF.** The treatments were inoculation with *G. diazotrophicus* and *H. seropedicae* (PGPB), bacteria and humic acid (PGPB+HA) or only humic acid (HA) isolated from vermicompost. The data were expressed in respect to the control treatment (%).

| Metabolite | HA | PGPB | PGPB+HA |
|---|---|---|---|
| 1,5-anhydroglucitol | −80[**] | −13[*] | 42[**] |
| 199177 carbohydrate | 58[ns] | 110[ns] | 187[ns] |
| 1-deoxyerythritol | 104[**] | 1,664[**] | 1,231[**] |
| 1-desoxypentitol | −28[ns] | 320[**] | 897[**] |
| 1-kestose | 79[ns] | 25[ns] | 74[ns] |
| 200391 disaccharide | −36[**] | 8[ns] | 81[**] |
| 200509 carbohydrate | 92[**] | −4[ns] | 54[**] |
| 200658 carbohydrate | −19[ns] | 28[ns] | 346[**] |
| 201952 carbohydrate | 46[ns] | 24[ns] | 271[**] |
| 202573 carbohydrate | 18[ns] | −1[ns] | 871[**] |
| 202832 disaccharide | −13[ns] | 21[ns] | 49[**] |
| 208658 carbohydrate | 19[ns] | 175[ns] | 779[ns] |
| 208845 carbohydrate | −19[**] | −27[**] | −48[**] |
| 214402 carbohydrate | −11[ns] | 202[ns] | 120[ns] |
| 225906 disaccharide | 33[ns] | 16[ns] | 83[ns] |
| 231180 carbohydrate | 61[ns] | 29[ns] | 240[*] |
| 231210 disaccharide | 64[ns] | 79[ns] | 182[ns] |
| 233289 carbohydrate | −19[ns] | 188[ns] | 911[**] |
| 238267 trisaccharide | 14[ns] | −5[ns] | −44[ns] |
| 267647 disaccharide | −8[ns] | 671[**] | 2169[**] |
| 285065 disaccharide | −25[ns] | 74[**] | 59[**] |
| 288331 trisaccharide | 55[ns] | 44[ns] | 100[ns] |
| 288365 trisaccharide | 12[ns] | 81[ns] | 168[ns] |
| 2-deoxyribose | 39[ns] | 35[ns] | 112[ns] |
| 2-ketoglucose dimethylacetal | −11[*] | 59[**] | −4[ns] |
| 2-ketogulonic acid | −32[ns] | −84[*] | −78[*] |
| 304945 carbohydrate | 58[**] | 66[**] | 15[ns] |
| 328803 carbohydrate | 14[ns] | 15[ns] | 79[**] |
| 400671 carbohydrate | 71[**] | −2[ns] | 50 |
| 424905 carbohydrate | −46[ns] | −69[*] | 179[**] |
| 506414 carbohydrate | 49[**] | 184[ns] | 323[**] |
| 553367 disaccharide | −52[**] | −37[**] | 43[**] |
| arabitol | −22[ns] | 65[ns] | 147[ns] |
| beta-glycerolphosphate | 17[ns] | 40[ns] | 146[ns] |
| beta-hexopyranose 1,6-anhydro | 122[ns] | 90[ns] | 157[*] |
| digitoxose | −25[ns] | −47[ns] | −35[ns] |
| enolpyruvate | 22[ns] | 85[ns] | 246[*] |
| erythritol | −2[ns] | 132[ns] | 169[*] |
| erythronic acid lactone | −18[ns] | 154[**] | 50[**] |
| erythrose | −30[ns] | 35[ns] | 65[ns] |
| fructose 1 phosphate | 101[ns] | 257[ns] | 738[**] |

**Table 1** (*continued*)

| Metabolite | HA | PGPB | PGPB+HA |
|---|---|---|---|
| fructose-6-phosphate | $-17^{ns}$ | $36^{ns}$ | $75^{ns}$ |
| fucose | $-51^{**}$ | $0^{ns}$ | $44^{**}$ |
| galactose | $160^{**}$ | $975^{**}$ | $1,876^{**}$ |
| galactose-6-phosphate 1 | $-5^{ns}$ | $27^{ns}$ | $107^{**}$ |
| glucoheptose | $-7^{ns}$ | $-5^{ns}$ | $107^{ns}$ |
| glucoheptulose | $7^{ns}$ | $152^{*}$ | $263^{**}$ |
| glucose | $85^{**}$ | $173^{**}$ | $299^{**}$ |
| glucose-1-phosphate | $38^{*}$ | $57^{**}$ | $116^{**}$ |
| glucose-6-phosphate 2 | $-21^{ns}$ | $129^{ns}$ | $413^{*}$ |
| glycero-guloheptose | $-13^{ns}$ | $-36^{ns}$ | $7,891^{ns}$ |
| hexose amino-2-deoxy | $-16^{ns}$ | $83^{ns}$ | $192^{**}$ |
| inulotriose | $120^{ns}$ | $-6^{ns}$ | $36^{ns}$ |
| isomaltose | $-77^{ns}$ | $95^{ns}$ | $586^{**}$ |
| lactobionic acid | $110^{**}$ | $152^{**}$ | $247^{**}$ |
| lactose | $510^{**}$ | $168^{ns}$ | $359^{**}$ |
| leucrose | $-66^{**}$ | $-30^{ns}$ | $-56^{**}$ |
| levanbiose | $107^{**}$ | $90^{**}$ | $116^{**}$ |
| levoglucosan | $-22^{ns}$ | $8^{ns}$ | $7^{ns}$ |
| maltose | $132^{*}$ | $-16^{ns}$ | $12^{ns}$ |
| maltotriose | $9^{ns}$ | $20^{ns}$ | $76^{ns}$ |
| mannose | $148^{**}$ | $22^{**}$ | $429^{**}$ |
| melezitose | $188^{ns}$ | $134^{ns}$ | $422^{ns}$ |
| melibiose | $-62^{**}$ | $943^{**}$ | $4,143^{**}$ |
| methylhexose | $45^{**}$ | $2^{ns}$ | $252^{**}$ |
| N-acetyl-D-hexosamine | $43^{**}$ | $37^{**}$ | $73^{**}$ |
| pentitol | $29^{ns}$ | $49^{ns}$ | $80^{ns}$ |
| propane-1,3-diol | $-8^{ns}$ | $2^{ns}$ | $15^{ns}$ |
| raffinose | $162^{**}$ | $-50^{**}$ | $16^{ns}$ |
| ribitol | $96^{ns}$ | $47^{ns}$ | $825^{**}$ |
| ribonic acid gamma-lactone | $-6^{ns}$ | $-14^{ns}$ | $92^{ns}$ |
| ribose | $548^{**}$ | $71^{ns}$ | $385^{**}$ |
| salicin | $211^{*}$ | $336^{**}$ | $1,782^{**}$ |
| sialicin | $179^{ns}$ | $-18^{ns}$ | $98^{ns}$ |
| sophorose | $49^{ns}$ | $127^{ns}$ | $297^{**}$ |
| sucrose-6-phosphate | $-42^{ns}$ | $79^{ns}$ | $185^{ns}$ |
| tagatose | $-24^{ns}$ | $22^{ns}$ | $175^{ns}$ |
| threitol | $-15^{ns}$ | $111^{ns}$ | $418^{ns}$ |
| trehalose | $-33^{ns}$ | $-17^{ns}$ | $133^{**}$ |
| furanose | $54^{ns}$ | $21^{ns}$ | $164^{ns}$ |
| xylitol | $91^{**}$ | $155^{**}$ | $146^{**}$ |
| xylonolactone | $41^{ns}$ | $32^{ns}$ | $71^{ns}$ |
| xylulose | $458^{**}$ | $228^{ns}$ | $1,264^{**}$ |

**Notes.**

ns, no significant.

$^{*}p < 0.05$.

$^{**}p < 0.01$.

**Table 2** **Amino acids identified on sugarcane methanolic leaves extracts by GC-MS TOF.** The treatments were inoculation with *G. diazotrophicus* and *H. seropedicae* (PGPB), bacteria and humic acid (PGPB+HA) or only humic acid (HA) isolated from vermicompost. The data were expressed in respect to the control treatment (%).

| Metabolite | HA | PGPB | PGPB+HA |
|---|---|---|---|
| 3-aminoisobutyric acid | $12^{ns}$ | $1^{ns}$ | $-20^{ns}$ |
| 4-hydroxyproline | $1^{ns}$ | $94^{**}$ | $369^{**}$ |
| 5-aminovaleric acid | $212^{**}$ | $24^{ns}$ | $64^{ns}$ |
| 5-hydroxynorvaline | $-26^{ns}$ | $154^{**}$ | $52^{ns}$ |
| asparagine | $-85^{**}$ | $177^{ns}$ | $937^{ns}$ |
| aspartic acid | $16^{ns}$ | $122^{**}$ | $629^{**}$ |
| β-alanine | $15^{ns}$ | $-21^{ns}$ | $0^{ns}$ |
| citrulline | $-18^{ns}$ | $359^{*}$ | $276^{ns}$ |
| cyano-L-alanine | $-3^{ns}$ | $155^{ns}$ | $186^{ns}$ |
| glutamic acid | $14^{ns}$ | $254^{**}$ | $250^{**}$ |
| glutamine | $-72^{ns}$ | $1,038^{**}$ | $595^{**}$ |
| glycine | $-14^{**}$ | $0^{ns}$ | $107^{**}$ |
| homoserine | $-38^{ns}$ | $216^{*}$ | $640^{**}$ |
| isoleucine | $45^{**}$ | $88^{**}$ | $135^{**}$ |
| lysine | $202^{ns}$ | $220^{ns}$ | $830^{ns}$ |
| methionine | $-42^{ns}$ | $161^{ns}$ | $199^{ns}$ |
| N-acetylornithine | $17^{ns}$ | $95^{ns}$ | $237^{ns}$ |
| N-hexanoylglycine | $184^{ns}$ | $404^{ns}$ | $646^{ns}$ |
| N-methylalanine | $-40^{ns}$ | $101^{**}$ | $99^{**}$ |
| O-acetylserine | $16^{ns}$ | $10^{ns}$ | $94^{ns}$ |
| oxoproline | $26^{**}$ | $192^{**}$ | $320^{**}$ |
| phenylalanine | $126^{ns}$ | $1,004^{**}$ | $819^{**}$ |
| proline | $278^{**}$ | $1,031^{**}$ | $1,155^{**}$ |
| serine | $-3^{ns}$ | $253^{**}$ | $81^{**}$ |
| threonine | $34^{ns}$ | $414^{**}$ | $101^{**}$ |
| thymine | $-15^{ns}$ | $216^{*}$ | $367^{**}$ |

**Notes.**

ns, no significant.

$^{*}p < 0.05$.

$^{**}p < 0.01$.

## Organic acids (Table 3)

All treatments induced the concentration of following acids: citric, alpha ketoglutaric, maleic, ribonic, oxalic, 2-deoxytetronic, lactic and idonic acid. The HA reduce the concentration of four organic acids (pyruvic acid $-61\%$; itaconic $-50\%$, mucic acid $-47\%$ and succinic acid, $-23\%$) and enhance the concentration of saccharic acid and isocitric lactone; PGPB decrease the concentration of only succinic acid ($-27\%$) and no organic acids significantly decreased in respect to the control in HA+PGPB treatment. The inoculation enhanced the acids 2-hydroxyglutaric, aminomalonic, gluconic acid lactone, isothreonic, itaconic, mannonic, mucic and pantothenic. The acids pyruvic (29%), fumaric acid (126%), 2-oxogluconic acid (730%), hexuronic acid (305%), ascorbic acid (368%),

**Table 3  Organic acids identified on sugarcane methanolic leaves extracts by GC-MS TOF.** The treatments were inoculation with *G. diazotrophicus* and *H. seropedicae* (PGPB), bacteria and humic acid (PGPB+HA) or only humic acid (HA) isolated from vermicompost. The data were expressed in respect to the control treatment (%).

| Metabolite | HA | PGPB | PGPB+HA |
|---|---|---|---|
| 2,3-dihydroxybutanoic acid | $30^{ns}$ | $31^{ns}$ | $66^{ns}$ |
| 2,5-furandicarboxylic acid | $8^{ns}$ | $19^{ns}$ | $85^{ns}$ |
| 2-deoxyribonic acid | $-11^{ns}$ | $-11^{ns}$ | $28^{ns}$ |
| 2-deoxytetronic acid | $95^{ns}$ | $95^{ns}$ | $100^{ns}$ |
| 2-hydroxyadipic acid | $68^{ns}$ | $7^{ns}$ | $185^{ns}$ |
| 2-hydroxyglutaric acid | $262^{ns}$ | $77^{ns}$ | $57^{ns}$ |
| 2-ketoadipic acid | $-37^{ns}$ | $8^{ns}$ | $-36^{ns}$ |
| 2-oxogluconic acid | $65^{ns}$ | $264^{ns}$ | $730^{**}$ |
| 3-hydroxy-3-methylglutaric acid | $-23^{ns}$ | $-4^{ns}$ | $5^{ns}$ |
| 3-hydroxypropionic acid | $27^{ns}$ | $31^{ns}$ | $87^{**}$ |
| 4-hydroxybutyric acid | $27^{ns}$ | $25^{ns}$ | $50^{ns}$ |
| 5-hydroxymethyl-2-furoic acid | $-12^{ns}$ | $-3^{ns}$ | $33^{ns}$ |
| adipic acid | $8^{ns}$ | $8^{ns}$ | $26^{ns}$ |
| alpha ketoglutaric acid | $7^{*}$ | $22^{*}$ | $83^{**}$ |
| aminomalonic acid | $36^{ns}$ | $116^{ns}$ | $83^{ns}$ |
| ascorbic acid | $114^{ns}$ | $155^{ns}$ | $368^{**}$ |
| citramalic acid | $-8^{ns}$ | $-27^{ns}$ | $-49^{ns}$ |
| citric acid | $98^{**}$ | $416^{**}$ | $748^{**}$ |
| dihydroxymalonic acid | $-72^{ns}$ | $0^{ns}$ | $298^{**}$ |
| fumaric acid | $2^{ns}$ | $-47^{ns}$ | $126^{**}$ |
| galactonic acid | $-39^{ns}$ | $34^{ns}$ | $37^{ns}$ |
| gluconic acid | $4^{ns}$ | $34^{ns}$ | $77^{ns}$ |
| gluconic acid lactone | $105^{ns}$ | $330^{**}$ | $412^{**}$ |
| glutaric acid | $12^{ns}$ | $179^{ns}$ | $184^{ns}$ |
| glycolic acid | $-16^{ns}$ | $1^{ns}$ | $19^{ns}$ |
| hexuronic acid | $-9^{ns}$ | $85^{ns}$ | $305^{*}$ |
| idonic acid | $150^{*}$ | $150^{*}$ | $268^{**}$ |
| isocitric lactone | $145^{**}$ | $10^{ns}$ | $140^{**}$ |
| isothreonic acid | $30^{ns}$ | $103^{**}$ | $211^{**}$ |
| itaconic acid | $-50^{**}$ | $31^{**}$ | $28^{**}$ |
| lactic acid | $76^{**}$ | $22^{**}$ | $43^{**}$ |
| maleic acid | $305^{**}$ | $263^{**}$ | $122^{**}$ |
| mannonic acid | $27^{ns}$ | $160^{ns}$ | $173^{ns}$ |
| methylmaleic acid | $-11^{ns}$ | $44^{ns}$ | $60^{ns}$ |
| mucic acid | $-47^{**}$ | $127^{**}$ | $408^{**}$ |
| oxalic acid | $745^{**}$ | $18^{**}$ | $259^{**}$ |
| pantothenic acid | $11^{ns}$ | $81^{**}$ | $307^{**}$ |
| propane-1,2,3-tricarboxylate /carballylic acid | $80^{ns}$ | $55^{ns}$ | $174^{ns}$ |

**Table 3** (*continued*)

| Metabolite | HA | PGPB | PGPB+HA |
|---|---|---|---|
| pyruvic acid | $-61^{**}$ | $0^{ns}$ | $29^{**}$ |
| ribonic acid | $33^{*}$ | $46^{**}$ | $155^{**}$ |
| saccharic acid | $133^{**}$ | $-13^{ns}$ | $235^{**}$ |
| salicylic acid | $-26^{ns}$ | $44^{ns}$ | $72^{*}$ |
| succinate semialdehyde/4-oxobutanoic acid | $-4^{ns}$ | $31^{ns}$ | $48^{ns}$ |
| succinic acid | $-23^{*}$ | $-27^{**}$ | $-15^{ns}$ |

Notes.

ns, no significant.

$^{*}p < 0.05$.

$^{**}p < 0.01$.

3-hydroxypropionic acid (87%), 2-hydroxyadipic acid (292%) were found in higher concentration in respect to control plants in the PGPB+HA treatment.

## Nitrogenous compounds (Table 4)

The nucleosides were the most nitrogenous compounds other than amino acids produce in large amount. Adenine was induced by all treatments and as also others nitrogenous compounds like ethanolamine, putrescine and N-acetyl-D-hexosamine. The HA treatments reduce the concentration of thymide ($-56\%$) while enhance 4-hydroxyquinoline-2-carboxylic acid ($+650\%$), 1,3-diaminopropane ($+131\%$), 2-deoxyadenosine ($+736\%$). The PGPB enhance 2-deoxyguanosine ($+1,179\%$), adenine ($+3,817\%$), guanine ($+777\%$) and cytosin (368%), butirolactan ($+800\%$), isonicotic acid ($+260\%$), thymidine ($+125\%$), thymine (216%) and uracil ($+213\%$), while PGPB+HA induce the carnitine ($+384\%$), maleimide ($+527\%$), orotic acid ($+214\%$), spermidine ($+749\%$), and 5′-deoxy-5′-methylthioadenosine ($+157\%$).

## Lipids (Table 5)

Four lipids were inhibited by HA in respect to control including the fatty acids palmitic, 2-monoolein and 6-hydroxycaproate and the plant sterol phytol while monoolein was inhibit by PGPB+HA. The 2-monoolein, lauric, methlylhexadecanoic and perlagonic acids and β-sistosterol were found in larger concentration in all treatments. The 6-hydroxycaproate dimer, dihydrosphingosine, 2-monoolein and palmitic acid were enhanced by inoculation while monoolein, behenic acid, eicosanoic and capric acid and cholesterol by HA. The following lipids were found in larger concentration only in PGPB+HA treatment: 2-monopalmiitn,1-monopalmiteolin, cerotic acid, linoleic mand linolenic acid, myristic acid, palmitoleic acid, pimelic acid, and the plant sterols phytol and the terpene squalene.

## Aromatic compounds (Table 6)

Two aromatics compounds decrease its concentration (pipecoluc acid and coniferin) due HA treatment while PGPB+HA reduce the level of 4-hydroxycinnamic acid. The concentration of shikimic acid, caffeic acid, 3,4-hydroxybenzoic acid and 2,3-dihydroxybenzoic acid were enhanced by all treatments. The inoculation induces the following compounds: chlorogenic, pipecolic and phtahlic acid while HA enhance the concentration of arbutin and 4-hydroxycinnamic acid. The PGPB+HA enhance

**Table 4  Nitrogenous (other than amino acids) identified on sugarcane methanolic leaves extracts by GC-MS TOF.** The treatments were inoculation with *G. diazotrophicus* and *H. seropedicae* (PGPB), bacteria and humic acid (PGPB+HA) or only humic acid (HA) isolated from vermicompost. The data were expressed in respect to the control treatment (%).

| Nitrogenous (other than amino acids) | HA | PGPB | PGPB+HA |
|---|---|---|---|
| 1,3-diaminopropane | 131$^*$ | 47$^{ns}$ | 196$^{**}$ |
| 2-deoxyadenosine | 736$^*$ | −14$^{ns}$ | 65$^{**}$ |
| 2′-deoxyguanosine | 108$^{ns}$ | 1,179$^{**}$ | 997$^{**}$ |
| 3,6-dihydro-3,6-dimethyl-2,5-bishydroxypyrazine | 4$^{ns}$ | −13$^{ns}$ | −1$^{ns}$ |
| 3-hydroxypyridine | 20$^{ns}$ | 175$^{ns}$ | 356$^{**}$ |
| 4-hydroxyquinoline-2-carboxylic acid | 650$^{**}$ | −40$^{ns}$ | 255$^{**}$ |
| 5′-deoxy-5′-methylthioadenosine | 4$^{ns}$ | 56$^{ns}$ | 157$^{**}$ |
| 5-methylcytosine | 31$^{ns}$ | 349$^{ns}$ | 503$^*$ |
| 6-hydroxynicotinic acid | 35$^{ns}$ | 26$^{ns}$ | 21$^{ns}$ |
| adenine | 377$^{**}$ | 3,817$^{**}$ | 4,971$^{**}$ |
| adenosine | −77$^{**}$ | 226$^{**}$ | 116$^{**}$ |
| biuret | −4$^{ns}$ | 72$^{ns}$ | 94$^{ns}$ |
| butyrolactam | −4$^{ns}$ | 800$^{**}$ | 708$^{**}$ |
| carnitine | 10$^{ns}$ | −10$^{ns}$ | 384$^{**}$ |
| cytidine | 15$^{ns}$ | 19$^{ns}$ | 151$^{ns}$ |
| cytosin | 5$^{ns}$ | 368$^*$ | 517$^{**}$ |
| ethanolamine | 192$^{**}$ | 86$^{**}$ | 94$^{**}$ |
| guanine | −50$^{ns}$ | 777$^{ns}$ | 2,119$^*$ |
| guanosine | 14$^{ns}$ | 1$^{ns}$ | 19$^{ns}$ |
| inosine | 52$^{ns}$ | −19$^{ns}$ | 12$^{ns}$ |
| isonicotinic acid | 34$^{ns}$ | 260$^{**}$ | 544$^{**}$ |
| maleimide | −16$^{ns}$ | 60$^{ns}$ | 527$^{**}$ |
| nicotinamide | 34$^{ns}$ | 17$^{ns}$ | 57$^{ns}$ |
| orotic acid | 19$^{ns}$ | 47$^{ns}$ | 214$^{**}$ |
| putrescine | 75$^{**}$ | 123$^{**}$ | 405$^{**}$ |
| pyrrole-2-carboxylic acid | 25$^{ns}$ | 30$^{ns}$ | 108$^{ns}$ |
| spermidine 2 | 4$^{ns}$ | 106$^{ns}$ | 749$^*$ |
| taurine | 16 | −3 | 21 |
| thymidine | −56$^{**}$ | 125$^{**}$ | 247$^{**}$ |
| uracil | −30$^{ns}$ | 213$^{**}$ | 81$^{**}$ |
| xanthine | −9$^{ns}$ | −25$^{ns}$ | 114$^{ns}$ |
| xanthosine | 11$^{ns}$ | −21$^{ns}$ | −5$^{ns}$ |
| xanthurenic acid | 7$^{ns}$ | 6$^{ns}$ | 63$^{ns}$ |

**Notes.**
ns, no significant.
$^*p < 0.05$.
$^{**}p < 0.01$.

**Table 5  Lipids identified on sugarcane methanolic leaves extracts by GC-MS TOF.** The treatments were inoculation with *G. diazotrophicus* and *H. seropedicae* (PGPB), bacteria and humic acid (PGPB+HA) or only humic acid (HA) isolated from vermicompost. The data were expressed in respect to the control treatment (%).

| Metabolite | HA | PGPB | PGPB+HA |
|---|---|---|---|
| 1-hexadecanol | 52[ns] | 17[ns] | 136[ns] |
| 1-monopalmiteolin | 9[ns] | −5[ns] | 123[ns] |
| 1-monopalmitin | 2[ns] | 30[ns] | 120[ns] |
| 223494 fatty acid methyl ester | 21[ns] | −16[ns] | 54[ns] |
| 2-hydroxyvaleric acid | −28[ns] | 7[ns] | 10[ns] |
| 2-monoolein | −59[**] | 25[*] | 43[**] |
| 2-monopalmitin | 113[ns] | −35[ns] | 357[**] |
| 2-monostearin | −4[ns] | 3[ns] | 168[*] |
| 2-palmitoleic acid | −22[ns] | −20[ns] | 133[ns] |
| 6-hydroxycaproate dimer | −82[**] | 103[**] | 134[**] |
| 6-hydroxycaproic acid | 49[ns] | 70[ns] | 99[ns] |
| arachidic acid | 9[ns] | −28[ns] | 4[ns] |
| azelaic acid | −36[ns] | 14[ns] | 105[ns] |
| behenic acid | 193[**] | −36[ns] | 439[**] |
| beta-sitosterol | 460[**] | 191[**] | 765[**] |
| caffeic acid | 28[ns] | 37[ns] | 65[ns] |
| capric acid | 60[**] | 12[ns] | 47[*] |
| cerotic acid | 8[ns] | 46[ns] | 464[**] |
| cholesterol | 222[*] | −7[ns] | 342[**] |
| cis-4-decene-1,10-dioic acid | 142[*] | 106[ns] | 429[**] |
| dihydrosphingosine | 47[ns] | 210[*] | 692[**] |
| dodecane | 54[ns] | 12[ns] | 11[ns] |
| dodecanol | 8[ns] | 22[ns] | 7[ns] |
| eicosenoic acid | 167[**] | −37[ns] | 417[**] |
| gamma-tocopherol | 14[ns] | −1[ns] | 123[*] |
| isopalmitic acid | 18[ns] | 43[ns] | 55[ns] |
| lauric acid | 179[**] | 165[**] | 202[**] |
| lignoceric acid | −13[ns] | −61[**] | 61[**] |
| linoleic acid | 2[ns] | 76[**] | 286[**] |
| linoleic acid methyl ester | 8[ns] | 42[ns] | 99[**] |
| linolenic acid | −6[ns] | 51[ns] | 597[**] |
| methylhexadecanoic acid | 34[*] | 364[**] | 579[**] |
| monoolein | 43[**] | −7[*] | −41[**] |
| montanic acid | −3[ns] | −19[ns] | 15[ns] |
| myristic acid | 100[ns] | 20[ns] | 175[*] |
| octadecanol | −11[ns] | 0[ns] | −8[ns] |
| oleic acid | −8[ns] | 22[ns] | 92[ns] |
| palmitic acid | −11[**] | 9[**] | 32[**] |
| palmitoleic acid | 41[ns] | 71[ns] | 297[**] |
| pelargonic acid | 38[**] | 117[**] | 69[**] |

**Table 5** (*continued*)

| Metabolite | HA | PGPB | PGPB+HA |
|---|---|---|---|
| pentadecanoic acid monoacylglycerol ester | $-5^{ns}$ | $-2^{ns}$ | $94^{ns}$ |
| phytol | $-84^{**}$ | $17^{ns}$ | $454^{**}$ |
| pimelic acid | $-30^{ns}$ | $-37^{ns}$ | $104^{**}$ |
| squalene | $-43^{ns}$ | $5^{ns}$ | $124^{**}$ |
| stigmasterol | $-29^{ns}$ | $-2^{ns}$ | $76^{ns}$ |
| suberyl glycine | $-20^{ns}$ | $-19^{ns}$ | $39^{ns}$ |
| α-tocopherol | $8^{ns}$ | $2^{ns}$ | $49^{*}$ |

**Notes.**
ns, no significant.
$^{*}p < 0.05$.
$^{**}p < 0.01$.

the concentration of parabanic acid, 4-hydroxybenzoate, 4-hydroxybenzaldehyde and coniferin.

## Multivariate analysis

Figure 2 showed the linear combination of original variables. The two first principal components captured 95% of total variance. As the principal components are orthogonal, it was possible to observe the relationship among the treatments and variables by the scores and loading graphics (Fig. 2). The first principal component captured 73% of total variance and separated PGPB+HA from other treatments while PC2 captured 22% of total variance and grouped HA and PGPB+HA treatments. The main compounds responsible for segregate PGPB+HA from others in PC1 were those from lipids metabolism while the changes on shikimic pathway put together HA and HA+PGPB treatments in PC2 axis.

## DISCUSSION

CG-MS TOF is one of most powerful tools for metabolomics approach and allows the compounds identification about fivefold more than the sugarcane metabolomics previously reported by *Glassop et al. (2007)*. The effect of treatment on sugarcane leaf metabolites concentration will be discussed, considering the main biochemical class of compounds and then an integrated view is presented (Fig. 3).

The glucose and glucose-1-P concentrations increase in response of inoculation in comparison with controls plant. Glucose-1-P is a key intermediate in several major carbon anabolic fluxes, such as sucrose, starch and cellulose biosynthesis while Glucose-6-P was found in higher concentration only in PGPB+HA showing effect on oxidative pentose phosphate cycle. Therefore, we observed enhance of erythritol in PGBP+HA treatment. Erythritol is an important nutrient for $N_2$-fixing plant endosymbionts and larger concentration of fructose-1-P is compatible with the erythritol biosynthesis (*Barbier et al., 2014*). Lactobionic acid, a disaccharide formed from gluconic acid and galactose was induced by inoculation and may be formed also by oxidation of lactose. Galactose is a member of the raffinose family of oligosaccharides and is the major carbohydrate translocated in many plants. A high free concentration of galactose is toxic in leaves reducing cell wall loosening (*Cheung & Cleland, 1991*). The inhibition of β-galactosidases

**Table 6 Aromatics compounds identified on sugarcane methanolic leaves extracts.** The treatments were inoculation with *G. diazotrophicus* and *H. seropedicae* (PGPB), bacteria and humic acid (PGPB+HA) or only humic acid (HA) isolated from vermicompost. The data were expressed in respect to the control treatment (%).

| Metabolite | HA | PGPB | PGPB+HA |
|---|---|---|---|
| vanillic acid | $18^{ns}$ | $24^{ns}$ | $60^{*}$ |
| syringic acid | $46^{ns}$ | $-11^{ns}$ | $83^{*}$ |
| shikimic acid | $311^{**}$ | $22^{**}$ | $385^{**}$ |
| pyrogallol | $-9^{ns}$ | $39^{ns}$ | $64^{ns}$ |
| pyrazine 2,5-dihydroxy | $60^{ns}$ | $85^{ns}$ | $189^{ns}$ |
| pipecolic acid | $-77^{**}$ | $197^{**}$ | $450^{**}$ |
| p-hydroquinone | $29^{ns}$ | $77^{ns}$ | $122^{ns}$ |
| phthalic acid | $-29^{*}$ | $72^{**}$ | $117^{**}$ |
| phenylethylamine | $-63^{ns}$ | $-13^{ns}$ | $4^{ns}$ |
| phenylacetic acid | $2^{ns}$ | $-9^{ns}$ | $29^{ns}$ |
| parabanic acid | $13^{ns}$ | $62^{ns}$ | $137^{**}$ |
| oxamic acid | $69^{ns}$ | $50^{ns}$ | $92^{ns}$ |
| gentisic acid | $45^{ns}$ | $-1^{ns}$ | $35^{ns}$ |
| ferulic acid | $-28^{ns}$ | $59^{ns}$ | $175^{ns}$ |
| dihydroabietic acid | $-29^{ns}$ | $17^{ns}$ | $78^{ns}$ |
| coniferin | $-77^{*}$ | $-33^{ns}$ | $201^{**}$ |
| cis-caffeic acid | $294^{**}$ | $138^{*}$ | $494^{**}$ |
| chlorogenic acid | $-20^{ns}$ | $133^{*}$ | $-30^{ns}$ |
| arbutin | $204^{**}$ | $21^{ns}$ | $127^{**}$ |
| 4-hydroxyphenylacetic acid | $6^{ns}$ | $12^{ns}$ | $12^{ns}$ |
| 4-hydroxymandelic acid | $-45^{ns}$ | $17^{ns}$ | $-24^{ns}$ |
| 4-hydroxycinnamic acid | $100^{**}$ | $-6^{ns}$ | $-19^{**}$ |
| 4-hydroxybenzoate | $19^{ns}$ | $1^{ns}$ | $98^{**}$ |
| 4-hydroxybenzaldehyde | $63^{ns}$ | $-48^{ns}$ | $87^{**}$ |
| 3-hydroxybenzoic acid | $-5^{ns}$ | $27^{ns}$ | $-20^{ns}$ |
| 3,4-dihydroxybenzoic acid | $46^{**}$ | $58^{**}$ | $29^{**}$ |
| 2-phenylpropanol | $47^{ns}$ | $10^{ns}$ | $15^{ns}$ |
| 2-(4-hydroxyphenyl)ethanol | $-11^{ns}$ | $-12^{ns}$ | $108^{**}$ |
| 2,3-dihydroxypyridine | $1^{ns}$ | $-9^{ns}$ | $-3^{ns}$ |
| 2,3-dihydroxybenzoic acid | $12^{**}$ | $52^{**}$ | $80^{**}$ |
| 1,2,4-benzenetriol | $14^{ns}$ | $-9^{ns}$ | $39^{ns}$ |

**Notes.**
ns, no significant.
$^{*}p < 0.05$.
$^{**}p < 0.01$.

releases galactose from cell wall polysaccharides and could occur as a growth response induced by auxins (*Thorpe et al., 1999*). It is well documented that HA and PGPB have auxin-like activity. It was also found that a high level of lactose and lactobionic acid and this former compound were associated with the metabolic signature related to the high plant growth rate in *Arabidopsis* by *Meyer et al. (2007)*.

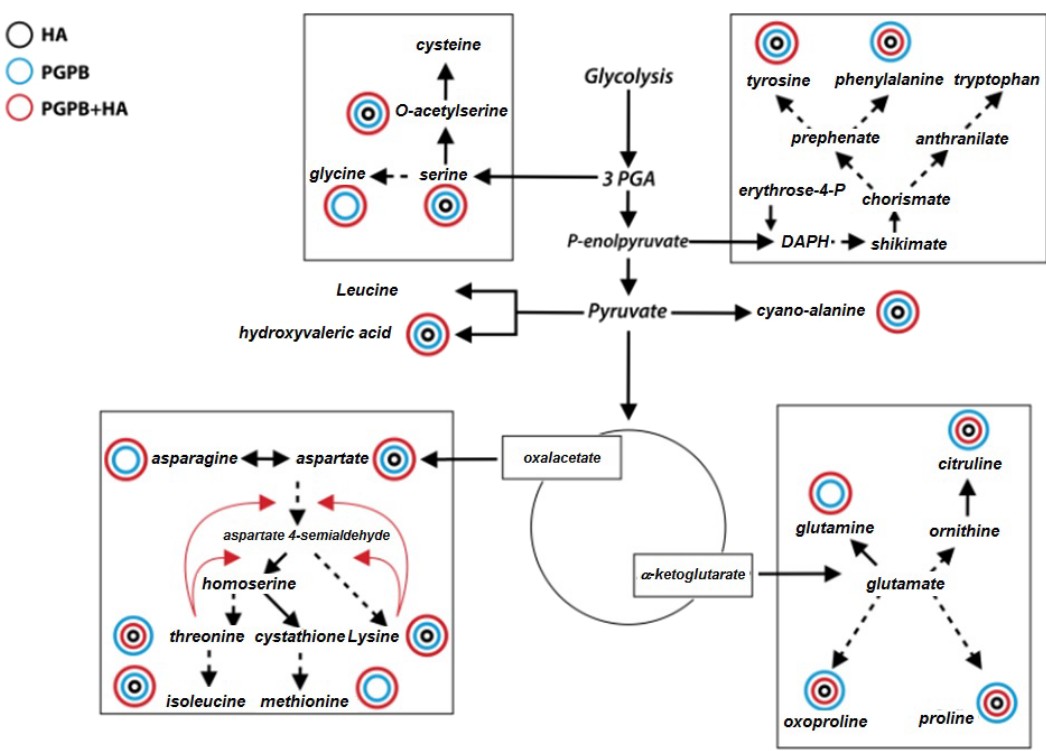

**Figure 1 Schematic representation of the effect of the treatments on the levels of amino acids in respect to control.** Schematic representation of the effect of the treatments on the levels of amino acids in respect to control. Rectangles containing aminoacids pools that derivates from pathways conversions of 3-phosphoglycerate (PGA), P-enolpyruvate, pyruvate, oxaloacetate and $\alpha$-ketoglutarate. Aminoacids with significant higher values related to control had shown color circles. The color circles represent the treatments (HA, black; PGPB, blue, HA+PGPB, red) and the magnitude of influence of each treatment in respect to control is represented by the size of the circle.

The effect of inoculation on plant carbohydrates was accentuated. In Gram negative bacteria, the inner core oligosaccharide typically contains residues of heptose (*Kosma, 2009*). The role of oligoliposaccharides in bacteria-host interaction is well known and described (*Nwodo, Green & Okoh, 2012*) and we found a huge increase of glycerol gluco heptose in plants treated with PGPB+HA. Another characteristic from the intensification of the carbohydrate metabolism by inoculation was the increase on aditol concentration in respect to control-plants (e.g., glucicol, ribitol, desoxypentiol, erythritol, Table 1). The alditols are products of photosynthesis and are more reduced, their corresponding sugars revealing metabolic boost induced by inoculation. Another intense change on the carbohydrate profile includes the production of osmoregulators compounds and sugars linked to cell wall rigidity. Melibiose, galactose, salicin and trehalose were previously described that their concentration enhances under abiotic or biotic stress (*Peña et al., 2004*). The larger concentration of mannose is compatible with high levels of galactose (*Herold & Lewis, 1977*). All treatments showed larger levels of salicin or 2-(hydroxymethyl) phenyl-*O*- β-ᴅ-glucopyranoside, one phenolic glycoside precursor of salicylic acid that have important roles in the ecological survival of plants against biotic and abiotic stress (*Mahdi,*

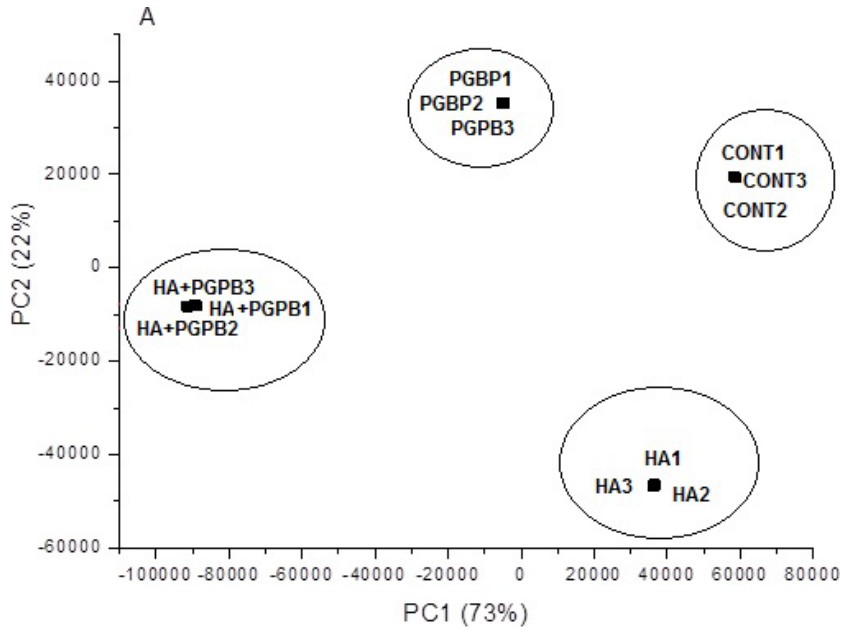

**Figure 2 PCA scores for CG-MS metabolic profile data.** PCA scores for CG-MS data indicating good separation of treatments in different groups according to bacteria inoculation (PGPB), humic acid (HA) application or its combined use (HA+PGPB) for metabolomic profiles, as well as good agreement among biological replicates. The loadings and their coefficient weight that indicate the position of the variables along the PCs and consequently their importance for that PC are shown in Supplemental Information 1.

*2014*). Finally, higher levels of hexosamines were found, a precursor of the synthesis of several lipids. It is not surprising, therefore, to find changes in the lipids profile induced by inoculation. In addition, as nitrogen metabolism is closely linked to carbohydrates synthesis, the amino acid profile was dramatically changed by the treatments compared with the control. The general effect of HA was the decrease in the concentration of most of amino acids in respect to control (Fig. 1). On the other hand, the cell concentration of acid amino acids was increased by PGPB including glutamine, asparagine and aspartic acid, the most abundant amino acids in leaves. The accurate determination of amino acids concentrations in the metabolome studies is a tricky question due massive number of compounds and some chemical transformation that may occur during the derivatization process. For instance, glutamine and glutamate can be partially converted to pyroglutamate during derivatization procedures as demonstrated by *Kanani & Klapa (2007)* and *Purwaha et al. (2014)*. However, in this work we use the data correction for such biases changes developed by the platform of primary metabolism identification from the Genome Center (UC-Davis, USA) which considers these possible changes (*Niehaus et al., 2017*).

Interestingly, an increase in alpha ketoglutarate derived from the citric acid cycle was noticed in all bacteria treatments that provides carbon skeleton for ammonium assimilation and biosynthesis of amino acids through the GS/GOGAT nitrogen assimilation pathway (EC 6.3.1.2 and EC 1.4.1.14, respectively). GS catalyzes the ATP-dependent assimilation of $NH_4^+$ into glutamine, using glutamate as a substrate, and it functions in a cycle with

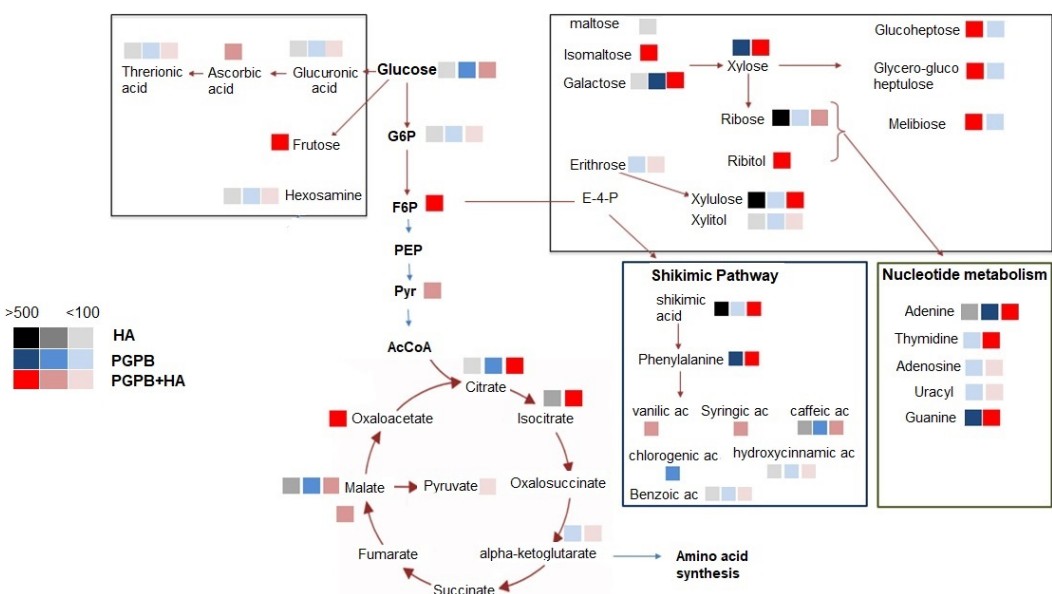

**Figure 3** **General view of the most important metabolites identified on main sugarcane biochemical pathways.** Representation of the most important metabolites identified on main sugarcane biochemical pathways. The color squares represent the treatments (HA, black; PGPB, blue; HA+PGPB: red) and the increase of concentration level (% in respect to control) was ranked according to the color-coded scale bar that range from 100 to 500× times concentration related to control plants.

GOGAT (glutamine-2-oxoglutarate aminotransferase), which catalyzes the reductive transfer of the amide group from glutamine to α-ketoglutarate (2-oxoglutarate), forming two molecules of glutamate. Modulation of GS activity in sugarcane leaves have been demonstrated by inoculation with diazotrophic bacteria and explain the partially nitrogen fixation abilities associated with sugarcane genotypes (*De Matos Nogueira et al., 2005*). In addition, changes in amino acid pools were previous demonstrated on sugarcane plants inoculated with diazotrophic bacteria that modulate nitrogen fixation and ammonium release by the bacteria (*Loiret et al., 2009*).

The proline and oxoproline concentration was larger in inoculate treatments, and the role of these amino acids in cell osmoregulation is well documented. The biotic stress induced by inoculation can promote a general plant response that includes increase of 5-hydroxy norvaline production and citrulline (*Yokota et al., 2002*). In a general view, plant inoculation sparks amino acids production especially those related to urea cycle (ornithine cycle). The 5-aminovaleric acid, isoleucine and proline were induced by HA. In plants, stress initiates a signal-transduction pathway in which increased cytosolic $Ca^{2+}$ activates $Ca^{2+}$/calmodulin-dependent glutamate decarboxylase activity and GABA synthesis. Elevated $H^+$ and substrate levels can also stimulate glutamate decarboxylase activity. GABA accumulation probably is mediated primarily by glutamate decarboxylase (*Shelp, Bown & Mclean, 1999*). *Ramos et al. (2015)* describe a mechanism involved $Ca^{2+}$ pulse uptake by plants stimulated by HA and describe a dynamic interaction between $Ca^{2+}$/$H^+$ efflux-influx as well as the induction on CDPK (calcium dependent phosphokinase) activity and $Ca^{2+}$ channels.

Furthermore, sugarcane transcriptomic study showed that protein kinases are main genetic response against drought and $N_2$-fixing inoculation (*Rocha et al., 2007*). It is possible to conclude based in metabolic profile analysis that the treatments improve the mechanism of adaptation based on ornithine cycle.

The inoculation enhances the synthesis of acidic amino acids and consequently it was possible to observe a significant enhance on nucleoside concentration (Tables 2 and 4). The interdependence of threonine, methionine and isoleucine metabolism in plants is well described *Joshi et al. (2010)*. Isoleucine is synthesized from threonine and methionine, which are derived from aspartate via enzymes located in the plastids and homoserine kinase (EC 2.7.1.39), which catalyzes the formation of O-phosphohomoserine from homoserine, leads to the formation of either threonine or methionine (*Hildebrandt et al., 2015*). PGPB and PGPB+HA enhance both the homoserine and isoleucine concentration.

The basic components of the citric acid cycle have their concentration enhanced by the treatments including oxalic, citric and α-ketoglutaric acid while succinic and fumaric acids decreased in respect to control (Table 3). High amounts of α-keto glutaric acid result in higher concentrations of glutaric acid and the glutamate/glutamine synthesis. This is coherent with the increase of ornithine, citrulline and proline and oxoproline production in plants inoculated. Another important item of evidence is the higher concentration of polyamines a direct product from ornithine transformation. Several intermediary products of TCA pathways were accumulated in higher concentration including dihydroxymalonic acid, isocitric lactone, 2-oxogluconic acid, succinate semialdehyde. Ascorbic acid was found in a high concentration. Ascorbate is a major metabolite in plants. It is an antioxidant and, in association with other components of the antioxidant system, protects plants against oxidative damage resulting from aerobic metabolism, photosynthesis and a range of pollutants (*Colville & Smirnoff, 2008*). This increase is compatible with the enhancement of the glucoronate metabolism observed (Table 3) and galactonic, gluconic, gluconic acid lactone, glutaric and glycolic acid concentrations since a biosynthetic pathway of ascorbic acid via GDP-mannose, GDP-L-galactose, L-galactose, and L-galactono-1,4-lactone has been proposed (*Smirnoff, 1996*). Pantothenic acid was also induced by inoculation and is a water-soluble vitamin (vitamin B5) essential for the synthesis of CoA and ACP, and a cofactor in energy yielding reactions including carbohydrate metabolism and fatty acid synthesis (*Coxon et al., 2005*).

Nucleoside and nucleoside derivatives (cytosine, guanine, adenine and thymine) participate in the bioenergetics process (ATP) and macromolecules synthesis including polysaccharides, phospholipids and glycolipids. The nucleosides synthesis was stimulated by treatments (Table 4). The growth cell acceleration must be preceded by carbohydrates metabolism and by the units necessary to build cell nuclei. The enhance of inosine concentration was observed in HA treatment. This compound is linked to cell growth activation. Furthermore, it was previously observed enhance of root growth of different plants with exogenous use of inosine (*Tokuhisa et al., 2010*). The HA like cytokinin activity was previously described (*Mora et al., 2012*) and the precursors of cytokinin was increased by HA treatment (3-hidroxyimethyl glutaric acid, oxoadenosine and inosine). Ethanolamine is a serine derivative (*Rontein et al., 2001*) and substrate to choline

synthesis and phosphatydiletanolamine and phosphadidilcoline, the main lipids in plant cell membranes.

The lipids metabolism was strongly affected by PGPB and responsible for the main differentiation among the other treatments according PCA analysis (Fig. 2, Table 5). The cytosine and 2-deoxyguanosine were huge increased with PGPB inoculation. The butyrolactam is synthesized from GABA cyclization by GABA-cysteine hydrogenase. This pathway may have been active by PGPB due the presence of high concentration of oxoproline. This amino acid is synthesized using glutamine-cysteine by cyclization catalyzed by (glutamiltransferase, EC 2.3.2.4) producing cysteine and oxoproline. This pathway was previously described in rice as a response to *Xanthomonas* infection (*Sana et al., 2010*). In the treatment with PGPB+HA one imide (maleimide) was found in a high concentration and it is a well-known cytotoxic compound. The main effect of maleimide is to block the vesicular transport, mainly the galactose transport (*Uemura et al., 2004*). Spermidine and putrescine were found in high concentrations with PGPB and PGPB+HA and are involved in cell growth inducing root growth and normally detected in high concentration in active tissue under stress (*Takahashi & Kakehi, 2009*). Previous report compares the action of HA with polyamines (*Young & Chen, 1997*; *Mora et al., 2012*). Larger concentrations of isonicotinic acid were found in the PGPB+HA treatment as well as the orotic acid, a precursor for the synthesis of water soluble B13 vitamin. The vitamins have important role in cell redox status and are enzymatic cofactor of metabolic reaction (*Asensi-Fabado & Munné-Bosch, 2010*).

The enhance of a range of cytosolic metabolites including glucose 6-P, malate derivatives, phosphoenolpyruvate and pyruvate support high rates of fatty acid synthesis by inoculated plants (*Pleite et al., 2005*). The long chain fatty acids as docosanoic acid (behenic acid), 2-monolein and hexadecanol were found in high concentration in HA treated plants. These long chain lipids are involved in membrane structure and dynamics regulating cell size but also the division and differentiation process (*Zheng, Rowland & Kunst, 2005*). Sphingolipids were found in larger concentration in HA and HA+PGPB plants. Elevated levels of complex sphingolipids were associated with cell apoptosis, terminal differentiation, or cell cycle arrest (*Du Granrut & Cacas, 2016*). In another way, the decrease on sphingolipids levels can lead to cell proliferation increase (*Bach & Faure, 2010*).

Sterols are synthesized by the mevalonate pathway, and it was possible to observe the enhancement of the sterols concentration induced by treatments in respect to the control. β-sitosterol is the main plant sterols involved into cellulose elongation chain and its increase were observed in HA and HA+PGPB treatments. Tocopherol is a precursor of main lipids soluble vitamin (vitamin E) and its concentration is regulated by biotic and abiotic stress (*Munne-Bosch, 2005*). The cholesterol concentration is often low in cell plants but the GC-MS TOF was sensible to detect significant variation in its concentration according the treatments. Phytol was abundant in HA+PGPB treatment and is the main precursor to brassinosteroids synthesis. Another lipid found in abundance in HA+PGPB was the terpene squalene ($C_{30}H_{50}$) precursor from major plant sterols (*Amarowicz, 2009*). The azelaic acid is a dicarboxylic acid with nine carbons and in the plant synthesis the even-numbered chains are privileged due to acetyl-coA mechanism reactions. This fatty

acid was present only in PGPB and HA+PGPB treatments. Previously it was found enhance in azelaic acid concentration in Arabidopsis infected by *Pseudomonas syringe* starting the plant defense against pathogen by activation of salicylic acid pathway (*Jung et al., 2009*).

In plants, the biosynthesis of isoprene units and common precursors for isoprenoid biosynthesis involves two distinct pathways. According to *Hemmerlin et al. (2006)*, the mevalonate (MVA) is utilized for the biosynthesis of non-plastidial isoprenoids (phytosterols, prenylated proteins, sesquiterpenoids), whereas plastidial isoprenoids (carotenoids, plastoquinone, diterpenes, monoterpenes, etc.) are synthesized via the alternative 2C-methyl-D-erythritol 4-phosphate (MEP) pathway. This pathway uses 1-Deoxy-D-xylulose 5-phosphate synthase (DXS) and thiamine to produce isoprenoids. We identified a significant enhancement in the xylulose (+1,263%) as well in the thymine (+4,970%) concentration in a clear indication of isoprenoids pathways stimulation by PGPB. Isoprenoids, represent the chemically and functionally most diversified class compounds including electron carriers (quinones), membrane constituents (sterols), vitamins (A, D, E and K), plant hormones (side chain of cytokinin, abscisic acid, gibberellins and brassinosteroids), and photosynthetic pigments (chlorophyll, phytol and carotenoids) (*Hemmerlin et al., 2006*). In fact, we observed an increase of *p*-hydroquinone, several sterols (β-sitosterol) phytol, tocopherol, stigmasterol, cholesterol cytosine, inosine and squalene in sugarcane treated with PGPB+HA. Furthermore, the accumulation of methylerytrol not discharge the enhance of isoprenoids products using a non-mevalonate pathway (MEP/DOXP) that use 2-methyl-D-erythritol 4-phosphate (MEP) and deoxy-xylulose 5 phosphate (DOXP) rote.

The shikimic pathway was also staggeringly changed by inoculation. Phenylpropanoids are a class of phenylalanine derivatives with a basic C6-C3 (phenyl-propane) skeleton (*Iriti & Faoro, 2009*). In turn, the essential amino acid phenylalanine arises from the shikimate pathway, as well as the other aromatic amino acids tyrosine and tryptophan. Precursors of this pathway are phosphoenolpyruvate, from glycolysis, and erythrose 4-phosphate from pentose phosphate (*Tzin & Galili, 2010*). Both compounds were its concentration enhanced by the treatments. Enolpiruvate enhance 22%, 85% and 246% with HA, PGPB and PGPB+HA treatment, respectively in respect to control while several derivatives of erytrose were found (erythritol, erythronic acid lactone, erythrose) in increased amount in inoculated plants. Then, we expected an influence on phenylpropanoids compounds, and it was confirmed by the influence on the multivariate analysis (PCA2, Fig. 3). The first step to the shikimic branch pathway is the accumulation of aromatic amino acids. An increase of phenylalanine concentration was also observed, and consequently the production of hydroxyl cinnamic derivatives. The main aromatic compounds induced by PGPB are derivatives of phenyl propanoids units as ferulic acids. The enhance of concentration on aromatics compound is often associated to improve redox status and antioxidative cell protection. Phenylalanine (tyrosine) ammonia-lyase (PAL/TAL; EC 4.3.1.5) catalyzes the first committed step in the biosynthesis of phenolics by converting phenylalanine to trans-cinnamic acid and tyrosine to p-coumaric acid. Phenylalanine usually is the preferred substrate, but the monocot enzyme can use both phenylalanine and tyrosine. *Schiavon et al. (2010)* observed enhance of PAL/TAL expression in maize treated with HA and therein the

concentration of total phenolic compounds. Here we provide an extensive list of aromatic compounds induce by HA. However, as the enhancement promoted in the aromatic amino acids phenylalanine and tyrosine by PGPB and PGPB+HA was larger, an increase in aromatic moieties was expected. Actually, a number of hydroxybenzoic acids were found including a direct derivative of salicylic acids like gentisic or quinic acids (caffeic acids) both involved in plant disease resistance. The improvement on the aromatic compounds synthesis induced by endophytic bacteria inoculation was previously related in sugarcane (*França et al., 2001*).

## CONCLUSION

Here we describe the changes in the metabolite profile in sugarcane inoculated with PGPB and HA. The glycolysis pathway was activated with high production of glycolysis and derivatives providing substrate to the pentose pathway and tricarboxylic acid cycle. The erithroses and heptuloses provide substrate to xylose and ribose synthesis and the nucleotides and acid shikimic acids was improved as well as the mevalonate pathway with the enhancement of phytosterols. The sugar amino acids were found in larger concentration as well as sphingolipids. Tricarboxylic acid cycle improvement also provides sources for lipids and amino acid metabolism, the main routes changed by inoculation. The cell growth acceleration was compatible with high nucleosides synthesis as well as water and lipid soluble vitamins. In general, several compounds that act as anti-oxidative agents and osmoprotectors were produced in inoculated plants, showing a rapid metabolic response to infection. In particular, the treatment with HA decreased the number of compounds with high cellular levels. This is compatible with previous proteome (*Carletti et al., 2008*) and transcriptome (*Trevisan et al., 2011*) studies that reported inhibition of differential expression for both protein and genes in response of HA. The HA increased the metabolic response of inoculation with PGPB. The levels of glucuronic acids and threonic acids increased when the plant was inoculated by PGPB+HA. Finally, analytical results obtained using GC-TOF MS clearly demonstrated the enhancement of TCA activity and showing the accumulation of α-Ketoglutaric acid, a central metabolic substrate to glutamine and glutamic acid synthesis precursor of amino acids ornithine, citrulline, proline, oxoproline, hydroxynorvaline and the polyamines (spermidine and putrescine). The level of fatty acids was strongly modified by PGPB+HA.

### Funding

This work was supported by Fundação de Amparo à Pesquisa do Estado do Rio de Janeiro (FAPERJ) and Conselho Nacional de Desenvolvimento Científico e Tecnológico (CNPq), INCT for Biological Nitrogen Fixation, Projeto FINEP-PLURICANA and International Foundation of Science (IFS). The funders had no role in study design, data collection and analysis, decision to publish, or preparation of the manuscript.

## Grant Disclosures

The following grant information was disclosed by the authors:

Fundação de Amparo à Pesquisa do Estado do Rio de Janeiro (FAPERJ).

Conselho Nacional de Pesquisa (CNPq).

Instituto Nacional de Ciência e Tecnologia para Fixação biológica de Nitrogênio (INCT-FBN).

Projeto FINEP-PLURICANA.

International Foundation of Science (IFS).

## Competing Interests

The authors declare there are no competing interests.

## Author Contributions

- Natalia O. Aguiar conceived and designed the experiments, performed the experiments, analyzed the data, prepared figures and/or tables, authored or reviewed drafts of the paper, approved the final draft.
- Fabio L. Olivares conceived and designed the experiments, analyzed the data, contributed reagents/materials/analysis tools, authored or reviewed drafts of the paper, approved the final draft.
- Etelvino H. Novotny analyzed the data, prepared figures and/or tables, authored or reviewed drafts of the paper, approved the final draft, supervised the statistical analysis.
- Luciano P. Canellas conceived and designed the experiments, performed the experiments, analyzed the data, contributed reagents/materials/analysis tools, prepared figures and/or tables, authored or reviewed drafts of the paper, approved the final draft.

## Data Availability

The raw data are provided in a Supplemental File.

## Supplemental Information

Supplemental information for this article can be found online at http://dx.doi.org/10.7717/peerj.5445#supplemental-information.

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

## FURTHER READING

**Aguiar NO, Medici L, Olivares F, Dobbss L, Torres-Netto A, Silva S, Novotny E, Canellas L. 2016.** Metabolic profile and antioxidant responses during drought stress recovery in sugarcane treated with humic acids and endophytic diazotrophic bacteria. *Annals of Applied Biology* **168**:203–213 DOI 10.1111/aab.12256.