# Peer review of "Changes in metabolic profiling of sugarcane leaves induced by endophytic diazotrophic bacteria and humic acids"

_PeerJ, doi:10.7717/peerj.5445_

## Round 0.1 · original submission · Major Revisions

Your manuscript has been reviewed by two experts in the field, and I agree with their comments and concerns. There are several issues with the presentation of your manuscript. Please address the reviewers’ concerns regarding clarity/meaning when revising the text. Please ensure the Introduction provides appropriate justification as to why the study is necessary. In addition, remove repetitive and/or redundant information from the Discussion. PeerJ does not provide a copy-editing service. Therefore, I request you have your manuscript corrected by a professional copy-editing service before resubmission, so that the important messages of your study can be properly conveyed to a wide readership.

In line with reviewer 2’s comments, please provide improved annotation information for the GC-MS data [including representative chromatogram(s)]. It is entirely possible for you to provide this as supplementary material so that reviewers/readers can determine the accuracy of your annotations.

Reviewer 1 ·

Basic reporting

no comment

Experimental design

no comment

Validity of the findings

no comment

Additional comments

This manuscript presents a study of the metabolic profiling of sugarcane leaves induced by endophytic diazotrophic bacteria and humic acids composting. Although the topic studied in the manuscript is interesting, the paper needs to improve in some aspects.
General Comment:
1. The experimental design of this interesting. However, the manuscript is not concise. Specifically, the Introduction section should be shortened and provide a better justification of why an investigation on metabolic profiling of sugarcane leaves is relevant and necessary. On the other hand, in my opinion, the Discussion section contains much repeated or redundant information.
2. The authors provided a schematic chart (like Fig 3), which is good. However, most discussions are not (sufficiently) conclusive. The authors do have a solid understanding of mechanisms but they need find a better way to present their thoughts.
Specific Comments:
1. The abstract better to be refined. Please introduce the aim and innovation of this study and bring to the abstract the better as possible of facts/data/description of each research objectives that were accomplished. Avoid being too general.
2. As the research defects described in this article are not prominent, leading to the innovation is not clear.
3. Please give a summary after each chapter.
4. Further refine the important conclusions of this paper.
5. Please provide details of three colors of cycle in Fig 1. The big or small cycle whether have different meaning in this figure.

Reviewer 2 ·

Basic reporting

The writing is clearly structured, but the text contains a number of sentences with error (e.g. missing words) and terms usually not found in a manuscript. An example would be the sentence 'However, the farm use of beneficial bacteria in non-leguminous is wispy compared with the rumbling success of soybean-rhizobia bioinoculants'. I get the meaning, but the clarity could be improved.

Data (integrals rather than raw data) is shared, but more information would improve it (see point 3).

The representation of the data needs quite a bit of improving in my opinion.

Tables: The changes in the table are expressed as % of control. Log fold change is a better value, as 400% looks more important than 25%, while there are indeed the same magnitude of change.

Figures: Figure 1 is confusing (what does the size of the circles mean, the caption 'Schematic representation of the effect of the treatments on the levels of amino acids in
respect to control' is not informative). Some of the labels are in Spanish. During glycolysis, GA3P is not converted to DHAP in a linear fashin, both can be interconverted by TPI, but it is the GA3P, which is converted to 3-PG (not 6-PG, as stated in the figure).

Similarly, Figure 3 is puzzling. What do the numbers on the legend mean? Why do some compounds have one square, others three and malate has four?

Experimental design

The method section (L163) states that the four treatments were carried out in four replicates. However, I can only see three replicates shown in the raw data. Do these three replicates correspond to three out of the four biological replicates?

Validity of the findings

I have my doubts about the GC-MS assignments, particularly for the carbohydrates. Carbohydrate assignment of GC-MS spectra is known to be tricky and I was particularly surprised to see that sucrose was missing from the list of annotated metabolites.

It would be better if raw data were reported with retention times and Validation and Quantification ions, which would make validating the assignments easier. Also, a representative chromatogramm, zoomed in on the most significant changes would be beneficial.

I am also suprised that the authors report glutamine and glutamate, as both are at least partially converted to pyroglutamate during derivaitisation.

Additional comments

Minor comments:
L116 - 1H NMR, not H1 NMR
L149 - nM, not ɳm

---

## Round 0.2 · Major Revisions

Your manuscript has been reviewed for a second time by an expert in metabolomics, who still has concerns regarding the reporting of raw data, retention times for reported compounds, and other issues that have not been addressed in your revised manuscript. Please address these issues. In addition, please acknowledge that partial conversion of glutamine and glutamate to pyroglutatmate occurs in MS approaches - the reviewer has kindly provided you with appropriate references in relation to this point. Your manuscript will be sent out for review again if you choose to submit a revised manuscript.

Reviewer 2 ·

Basic reporting

I think the reporting of the raw data still needs improving. While the authors have provided the raw data, no information about retention times for the reported compounds was given, neither about the quantitation and validation ions that are a feature of Fiehn library matching.

Despite the point about figure 1 and 3 being confusing and the authors' statement that those were modified, the figure legends were not modified according to the track changes document.

Experimental design

My point has been addressed.

Validity of the findings

My comments about the partial conversion of glutamine and glutamate to pyroglutamate (or oxo-proline as it also called) is not a knock on the Fiehn library, which a fantastic resource.

It is, however, an acknowledged fact that partial conversion happens in both GC-MS and LC-MS for these compounds (see, for example https://pubs.acs.org/doi/abs/10.1021/ac501451v for LC-MS and https://www.ncbi.nlm.nih.gov/pubmed/17052933 for GC-MS) and should therefore being acknowledged in the discussion

Another point concerns the sugars found in the experiment. While I can follow the major changes, some of the identified compounds seems to have no biological connection to sugar cane, e.g. salicin is found in willow bark and melezitose is a product or aphids and other insects. Are these confirmed components of sugar cane as well?

---

## Round 0.3 · accepted · Accept

Your manuscript has been reviewed, and is considered accepted for publication. The reviewer, however, has noted the RT values are not reported in the document. It would be useful if you could include the RT values if they are available to you, as this will allow readers to ascertain if a peak has been assigned multiple compound identities. If the RT values are not available, please add wording to this effect in the final version of the manuscript, with a clear explanation as to why the RT values are not included. Please indicate in your communications with the Editorial Office whether you have added the RT values or a statement as to why the RT values are not included. [# Staff note - this text can be added while in production #]

# Reviewer 2 ·

Basic reporting

There is still no RT in the raw data, but otherwise it is all reported. The RT would have been helpful to ascertain if a peak had been assigned multiple compound identities.

Experimental design

fine

Validity of the findings

fine, with caveats outlined above